# Distribution and associations for antimicrobial resistance and antibiotic resistance genes of *Escherichia coli* from musk deer (*Moschus berezovskii*) in Sichuan, China

**Hang Liu**[1☯], **Shulei Pan**[1☯], **Yuehong Cheng**[2☯], **Lijun Luo**[1], **Lei Zhou**[3], **Siping Fan**[1], **Liqin Wang**[4], **Shaoqi Jiang**[1], **Ziyao Zhou**[1], **Haifeng Liu**[1], **Shaqiu Zhang**[1], **Zhihua Ren**[1], **Xiaoping Ma**[1], **Suizhong Cao**[1], **Liuhong Shen**[1], **Ya Wang**[1], **Dongjie Cai**[1], **Liping Gou**[1], **Yi Geng**[1], **Guangneng Peng**[1], **Qigui Yan**[1], **Yan Luo**[1]*, **Zhijun Zhong**[1]*

1 College of Veterinary Medicine, Sichuan Agricultural University, Key Laboratory of Animal Disease and Human Health of Sichuan, Chengdu, China, 2 Sichuan Wolong National Natural Reserve Administration Bureau, Wenchuan, Sichuan, China, 3 Sichuan Institute of Musk Deer Breeding, Dujiangyan, China, 4 The Chengdu Zoo, Institute of Wild Animals, Chengdu, China

☯ These authors contributed equally to this work.

* zhongzhijun488@126.com (ZZ); lycjg@163.com (YL)

**Data Availability Statement:** All relevant data are within the paper and its Supporting information files.

## Abstract

This study aimed to investigate the antimicrobial resistance (AMR), antibiotic resistance genes (ARGs) and integrons in 157 *Escherichia coli* (*E. coli*) strains isolated from feces of captive musk deer from 2 farms (Dujiang Yan and Barkam) in Sichuan province. Result showed that 91.72% (144/157) strains were resistant to at least one antimicrobial and 24.20% (38/157) strains were multi-drug resistant (MDR). The antibiotics that most *E. coli* strains were resistant to was sulfamethoxazole (85.99%), followed by ampicillin (26.11%) and tetracycline (24.84%). We further detected 13 ARGs in the 157 *E. coli* strains, of which $bla_{TEM}$ had the highest occurrence (91.72%), followed by *aac(3')-lid* (60.51%) and $bla_{CTX-M}$ (16.56%). Doxycycline, chloramphenicol, and ceftriaxone resistance were strongly correlated with the presence of *tetB*, *floR* and $bla_{CTX-M}$, respectively. The strongest positive association among AMR phenotypes was ampicillin/cefuroxime sodium (OR, 828.000). The strongest positive association among 16 pairs of ARGs was *sul1*/*floR* (OR, 21.667). Nine pairs positive associations were observed between AMR phenotypes and corresponding resistance genes and the strongest association was observed for CHL/*floR* (OR, 301.167). Investigation of integrons revealed *intI1* and *intI2* genes were detected in 10.19% (16/157) and 1.27% (2/157) *E. coli* strains, respectively. Only one type of gene cassettes (*drA17-aadA5*) was detected in class 1 integron positive strains. Our data implied musk deer is a reservoir of ARGs and positive associations were common observed among *E. coli* strains carrying AMRs and ARGs.

## 1. Introduction

Musk deer is one of the first-class protected and endangered animals in China [1]. In captive environment, intestinal disease is the main disease that threatens musk deer breeding [2, 3].

**Funding:** This work was funded by the National Key Research and Development Program of China (2018YFD0500900, 2016YFD0501009), the Chengdu Giant Panda Breeding Research Foundation (CPF2017-05, CPF2015-4) and the Science and Technology Achievements Transfer Project in Sichuan province (2022JDZH0026).

**Competing interests:** The authors have declared that no competing interests exist.

Antimicrobial resistance (AMR) of *E. coli* has become a global challenge due to the wide usage of antibiotics in clinic [4]. In particular, *E. coli* isolates that were resistant to at least three or more categories of antibiotics are considered as multi-drug resistant (MDR) [5], which increased the difficulty of treating animal bacterial diseases and also a potential risk to public health.

Antibiotic resistance gene (AGR) was considered to be the main factor of antimicrobial resistance [6] and become a new contaminant for environmental pollution [7]. ARGs are usually carried by integrons, transposons and plasmids to mediate the horizontal transmission of AMR [8]. Integrons can capture, convert and adapt resistance gene cassettes into functionally expressed genes via self-efficient gene expression system [9]. Integron usually links to plasmid, which can transfer ARGs among various bacterial species [10].

*E. coli* is usually considered as an indicator organism to AMR [6, 11, 12]. Though AMR associated with ARGs have been reported in *E. coli* from different animals (honeybees, poultry and pigs) [13–15], antimicrobial resistance of *E. coli* in musk deer remains unknown. This study aimed to examine the antimicrobial resistance phenotype, antimicrobial resistance genes and integron in *E. coli* to better understand the association among AMR phenotypes and ARGs in musk deer.

## 2. Materials and methods

### 2.1 Sample collection

From Sep to Oct 2020, fresh fecal samples were collected from 157 clinically healthy musk deer in Dujiang Yan (n = 101) and Barkam (n = 56) (S1 Table). Each sample was collected from an individual. Fresh fecal specimens (approximately 10 g) from each musk deer were collected immediately by feeders after defecation on the ground and then quickly transferred into individual 50-mL plastic containers. After collection, samples were stored at 4°C in a cooler and transported to the laboratory for bacterial isolation within 12 h.

This study was reviewed and approved by the Institutional Animal Care and Use Committee of Sichuan Agricultural University under permit number DYY-2020103018. Prior to the collection of fecal specimens from captive musk deer, permission was obtained from Sichuan Institute of Musk Deer Breeding in Sichuan, China.

### 2.2 Isolation and enumeration of *E. coli*

Fecal samples were pre-enriched in buffered peptone water (Oxoid, UK) and incubated for 4–6 h at 37°C. All isolates were presumptively identified by Gram staining, MacConkey (Solarbio, BeiJing), and then confirmed by eosin methylene blue agar growth (Chromagar, France) (Solarbio, BeiJing), and then confirmed by eosin methylene blue agar growth (Chromagar, France) [16, 17] and biochemically identified by API 20E system (BioMerieux, France). The 16SrRNA (F: AGAGTTTGATCCTGGCTCAG, R: GGTTACCTTGTTACGACTT) of all strains were further amplified to confirm the isolates are *E. coli*.

### 2.3 Antimicrobial susceptibility testing

The susceptibilities of all isolates to 27 antimicrobials were tested using the standard disk diffusion method recommended by the Clinical and Laboratory Standards Institute [18]. The following antimicrobial disks (Oxoid) were used: aminoglycosides (gentamicin, CN, 10 μg; tobramycin, TOB, 10 μg; amikacin, AK, 30 μg; netilmicin, NET, 10 μg), chloram phenicols (chloramphenicol, CHL, 30 μg), quinolones (ciprofloxacin, CIP, 5 μg), sulfonamide (sulfamethoxazole, RL, 25 μg), tetracyclines (tetracycline, TE, 30 μg; doxycycline, DO, 30 μg;

**Table 1. The use of antibiotics in two farms, top phenotypic resistance and ARGs profile in *E. coli* (n = 157).**

| Sample source | The use of antibiotics | Top phenotypic resistance pattern (%) | Percentage of resistance isolates (%) | Percentage of MDR isolates (%) | Top ARGs pattern (%) | Percentage of integron positive isolates (%) |
|---|---|---|---|---|---|---|
| Dujiang Yan | AMC/CN/CRO/ENR/FFC | RL (47.5) | 94.1 | 19.8 | $bla_{TEM}/aac(3')$-$IId$ (29.7) | 8.9 |
| Barkam | AMC/CRO/FOX | RL (39.3) | 89.3 | 32.1 | $bla_{TEM}/aac(3')$-$IId$ (23.2) | 14.1 |

tigecycline, TGC, 15 μg; minocycline, MH, 30 μg), β-lactams (piperacillin/tazobactam10:1, TZP, 100 μg/10 μg; ertapenem, ETP, 10 μg; imipenem, IPM, 10 μg; meropenem, MEM, 10 μg; cephazolin, KZ, 30 μg; cefuroxime sodium, CXM, 30 μg; cefotaxime, CTX, 30 μg, ceftriaxone, CRO, 30 μg; cefepime, FEP, 30 μg; cefoxitin, FOX, 30 μg, aztreonam, ATM, 30 μg; ampicillin, AMP, 10 μg; amoxicillin/clavulanic acid 2:1, AMC, 20/10 μg; ampicillin/sulbactam 1:1, SAM, 10/10 μg; fosfomycin, FOS, 50 μg). Results were interpreted in accordance with CLSI criteria. *E. coli* ATCC25922 was used as a control. The 27 antibiotics were chosen with the following rationale: AMC, CN, CRO, AK, ENR, FFC and FOX have been used in two farms (Table 1); *E. coli* in other *Cervidae* have been found resistant to TOB, NET, CHL, CIP, RL, TE, TGC, MH, TZP, MEM, KZ, CXM, CTX, CAZ, FEP and SAM [19, 20]; AML, AMP, TET, DOX and RL have been widely used in animal husbandry [21]. MDR refers to the concurrent resistance of three or more categories of antibiotics [22].

## 2.4 DNA extraction and screening for antimicrobial resistance genes and integrons

Total *E. coli* genomic DNA was extracted by kit according the manufacturer (Tiangen Biotech, Beijing, China). Extracted DNA was stored at -20˚C. Thirteen different resistance genes and *intl 1*, *intl 2* and *intl 3* gene amplified by polymerase chain reaction (PCR). Subsequently, strains harboring *intl 1* or *intl 2* genes were elaborated for the amplification of variable regions (VRs). The PCR products were separated by gel electrophoresis in a 1.5% agarosegel stained with GoldView™ (Sangon Biotech, Shanghai, China), visualized under ultraviolet light and photographed using a gel documentation system (Bio-Rad, Hercules, USA). Primers and detailed conditions for PCR are described in S2 Table. The PCR reaction mixture was prepared with a composition of 12.5 μL of master mix (Origin, India), 2 μL template DNA, 1 μL primers (forward and reverse) and the volume raised to 25 μL with the water free from nuclease.

## 2.5 Sequence and data analysis

All positive PCR products were directly sequenced by BGI (Beijing, China) in both directions. Sequences were analyzed online using BLAST (http://blast.ncbi.nlm.nih.gov). P-values < 0.05 were considered to be statistically significant. The association between AMR phenotypes and the ARG was calculated, and the association was considered significant at a P-value of < 0.05 and was reported as an odds ratio (OR) with 95% confidence intervals (CI). An OR of > 1 was considered as positive association or the increasing probability of the cooccurrence of the genotype or phenotype, while an OR of < 1 was considered as negative association or the decreasing probability of the cooccurrence of the genotype or phenotype. All analyses were conducted using the SPSS 27 software (StataCorp Lp, College Station, TX, USA).

## 3. Results

### 3.1 Antimicrobial susceptibility testing for 157 *E. coli*

The distribution of antibiotic resistant in 157 *E. coli* was summarized in S1 Table. One hundred and forty-four isolates (91.71%, 144/157) were resistant to at least one antibiotic. Most of these isolates showed resistance to RL (85.99%, 135/157), AMP (26.11%, 41/157), TE (24.84%, 39/157), CTX (24.20%, 38/157), CXM (23.57%, 37/157) and KZ (22.93%, 36/157). All strains were sensitive to AK, ETP, IPM, MEM and FOX. Fifty-five antibiotic resistance patterns were observed, and the top three prevalent patterns are RL (44.59%, 70 isolates), RL/TE (4.46%, 7 isolates) and RL/TE/DO (4.46%, 7 isolates). Thirty-eight isolates (24.20%, 38/157) were found MDR in our study (Dujiang Yan, 19.80%, 20/101; Barkam, 32.14%, 18/56). Thirteen strains were resistant to more than 10 kinds of antibiotics, DJY62 and DJY75 were resistant to 15 antibiotics.

The details of AMRs for *E. coli* strains from captive musk deer in two farms were shown in Table 2 and Fig 1. Both Dujiang Yan (93.07%, 94/101) and Barkam (89.27%, 50/56) have high

**Table 2. The distribution among *E. coli* resistance phenotype from captive musk deer between Dujiang Yan (n = 101) and Barkam (n = 56).**

| Antibiotics | Dujiang Yan(n = 101) | | Barkam(n = 56) | | P-value | Total(n = 157) | |
|---|---|---|---|---|---|---|---|
| | Number of resistance strains | Resistance rate (%) | Number of resistance strains | Resistance rate (%) | | Number of resistance strains | Resistance rate (%) |
| CN | 6 | 5.94 | 4 | 7.14 | 0.745 | 10 | 6.37 |
| TOB | 3 | 2.97 | 2 | 3.57 | 1.000 | 5 | 3.18 |
| AK | 0 | 0.00 | 0 | 0.00 | - | 0 | 0.00 |
| NET | 2 | 1.98 | 0 | 0.00 | 0.538 | 2 | 1.27 |
| TE | 19 | 18.81 | 20 | 35.71 | 0.022 | 39 | 24.84 |
| DO | 18 | 17.82 | 12 | 21.43 | 0.673 | 30 | 19.11 |
| TGC | 1 | 0.99 | 0 | 0.00 | 1.000 | 1 | 0.64 |
| MH | 4 | 3.96 | 5 | 8.93 | 0.283 | 9 | 5.73 |
| CHL | 9 | 8.91 | 7 | 12.50 | 0.583 | 16 | 10.19 |
| TZP | 2 | 1.98 | 1 | 1.79 | 1.000 | 3 | 1.91 |
| ETP | 0 | 0.00 | 0 | 0.00 | - | 0 | 0.00 |
| IPM | 0 | 0.00 | 0 | 0.00 | - | 0 | 0.00 |
| MEM | 0 | 0.00 | 0 | 0.00 | - | 0 | 0.00 |
| KZ | 21 | 20.79 | 18 | 32.14 | 0.049 | 36 | 22.93 |
| CXM | 18 | 17.82 | 19 | 33.93 | 0.031 | 37 | 23.57 |
| CTX | 22 | 21.78 | 16 | 28.57 | 0.437 | 38 | 24.20 |
| CRO | 18 | 17.82 | 16 | 28.57 | 0.156 | 34 | 21.66 |
| CAZ | 8 | 7.92 | 6 | 10.71 | 0.569 | 14 | 8.92 |
| FEP | 8 | 7.92 | 7 | 12.50 | 0.400 | 15 | 9.55 |
| FOX | 0 | 0.00 | 0 | 0.00 | - | 0 | 0.00 |
| ATM | 13 | 12.87 | 10 | 17.86 | 0.481 | 23 | 14.65 |
| AMP | 21 | 20.79 | 20 | 35.71 | 0.057 | 41 | 26.11 |
| AMC | 0 | 0.00 | 1 | 1.79 | 0.357 | 1 | 0.64 |
| SAM | 0 | 0.00 | 3 | 5.36 | 0.044 | 3 | 1.91 |
| CIP | 9 | 8.91 | 16 | 28.57 | 0.002 | 25 | 15.92 |
| RL | 86 | 85.15 | 49 | 87.50 | 1.000 | 135 | 85.99 |
| FOS | 0 | 0.00 | 2 | 3.57 | 0.126 | 2 | 1.27 |

Note: -, indicates no results available.

P-value: $P < 0.05$ was considered statistically significant between Dujiang Yan and Barkam.

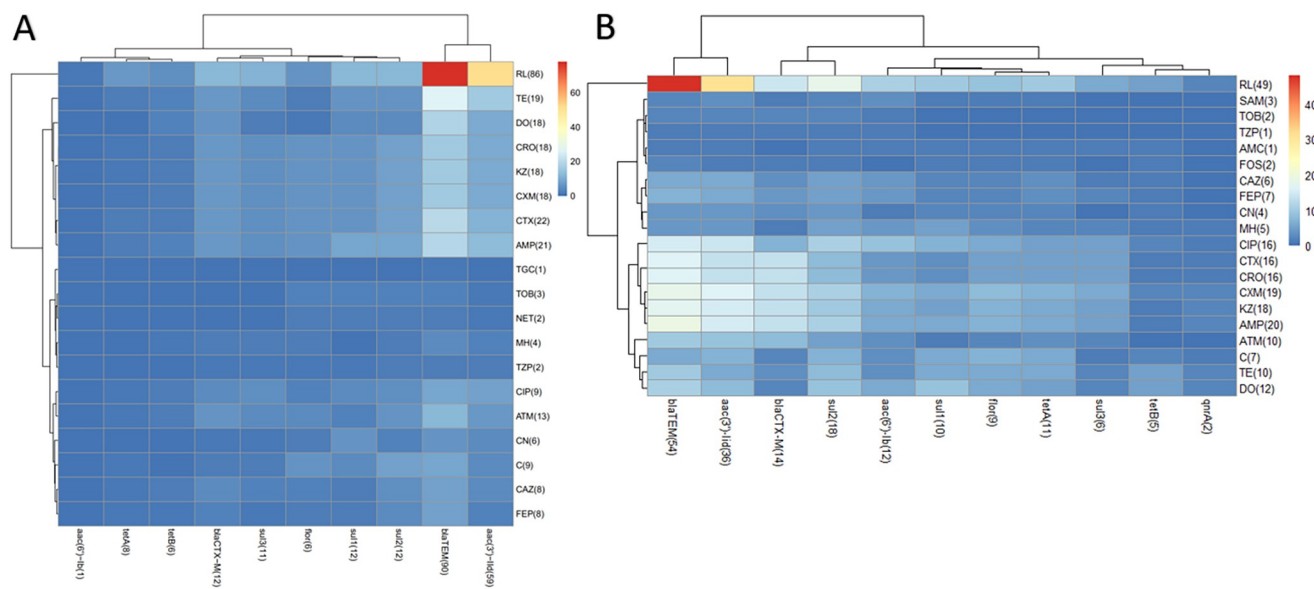

**Fig 1. Heat map demonstrating the distribution of ARGs and AMRs in *E. coli* isolates from musk deer in Sichuan, China.** The color scale on the right of figure showed the total number of *E. coli* isolates, in each of which carried a pair of antibiotic resistance genes and antibiotic corresponding to abscissa and ordinate, blue indicating the number is low, white indicating intermediate value, and red indicating the number is high. The number of positive isolates were also listed in the figure. A: Eighty-six *E. coli* strains were resistant to RL and the $bla_{TEM}$ was the most prevalent genes among all tested genes in Dujiang Yan. B: The results were similar in Dujiang Yan, the most prevalent AMR was RL and $bla_{TEM}$ was the most prevalent genes among all tested genes in Barkam.

rate of antibiotic resistance and showed no significant difference ($P > 0.05$). Significant differences in resistance to TE, KZ, CXM, SAM and CIP were observed from Barkam and Dujiang Yan ($P < 0.05$).

## 3.2 Distribution of 13 antibiotic resistance genes (ARGs) for 157 *E. coli*

As S1 Table showed, at least one of the ARGs were detected in 149 (149/157, 94.90%) strains. Among the 13 ARGs, 11 were detected: $bla_{TEM}$ (91.72%), *aac(3')-IId* (60.51%), *sul2* (19.11%), $bla_{CTX-M}$ (16.56%), *sul1* (14.01%), *tetA* (12.10%), *sul3* (10.83%), *floR* (9.55%), *aac(6')-Ib* (8.28%), *tetB* (7.01%), *qnrA* (1.27%). The remaining 2 ARGs ($bla_{SHV}$ and *catA1*) were not detected. Fifty ARGs patterns were observed, of which $bla_{TEM}$ /*aac(3')-IId* (27.39%, 43/157) was the most prevalent pattern. It is worth noting that 20 isolates carried more than 5 types of ARGs. Strain Bar35 from Barkam carried the most ARGs (10 types).

The details of ARGs carrying rate for *E. coli* strains from captive musk deer in two farms were shown in Tables 3 and 4, Fig 1. Among strains that carried ARGs, 94 (93.07%, 94/101) strains were isolated from Dujiang Yan and 55 (98.21%, 55/56) strains were isolated from Barkam. The most prevalent ARGs pattern was $bla_{TEM}$ /*aac(3')-IId* in Dujiang Yan (29.70%, 30/101) and in Barkam (23.21%, 13/56), respectively. The ARGs carrying rate in Barkam was 98.21%, higher than that in Dujiang Yan (93.07%). In addition, $bla_{SHV}$, *catA1* and *qnrA* were not detected in Dujiang Yan, $bla_{SHV}$ and *catA1* were not detected in Barkam. Significant difference in $bla_{CTX-M}$, *floR*, *tetA*, *sul2*, *aac(6')-Ib* were observed between strains from Barkam and Dujiang Yan ($P < 0.05$).

**Table 3. The distribution of antibiotic resistance genes among *E. coli* from captive musk deer between Dujiang Yan (n = 101) and Barkam (n = 56).**

| ARGs | Dujiang Yan(n = 101) | | | Barkam(n = 56) | | | P-value | Total(n = 157) | | |
|---|---|---|---|---|---|---|---|---|---|---|
| | Number of positive strains | Number of negative strains | Positive rate (%) | Number of positive strains | Number of negative strains | Positive rate (%) | | Number of positive strains | Number of negative strains | Positive rate (%) |
| *bla*$_{TEM}$ | 90 | 11 | 89.11 | 54 | 2 | 96.43 | 0.138 | 144 | 13 | 91.72 |
| *bla*$_{CTX-M}$ | 12 | 89 | 11.88 | 14 | 42 | 25.00 | 0.044 | 26 | 131 | 16.56 |
| *bla*$_{SHV}$ | 0 | 101 | 0.00 | 0 | 56 | 0.00 | - | 0 | 157 | 0.00 |
| *catA1* | 0 | 101 | 0.00 | 0 | 56 | 0.00 | - | 0 | 157 | 0.00 |
| *flor* | 6 | 95 | 5.94 | 9 | 47 | 16.07 | 0.049 | 15 | 142 | 9.55 |
| *tetA* | 8 | 93 | 7.92 | 11 | 45 | 19.64 | 0.041 | 19 | 138 | 12.10 |
| *tetB* | 6 | 95 | 5.94 | 5 | 51 | 8.93 | 0.523 | 11 | 146 | 7.01 |
| *sul1* | 12 | 89 | 11.88 | 10 | 46 | 17.86 | 0.341 | 22 | 135 | 14.01 |
| *sul2* | 12 | 89 | 11.88 | 18 | 38 | 32.14 | 0.003 | 30 | 127 | 19.11 |
| *sul3* | 11 | 90 | 10.89 | 6 | 50 | 10.71 | 1.000 | 17 | 140 | 10.83 |
| *qnrA* | 0 | 101 | 0.00 | 2 | 54 | 3.57 | 0.126 | 2 | 155 | 1.27 |
| *aac(3')-IId* | 59 | 42 | 58.42 | 36 | 20 | 64.29 | 0.500 | 95 | 62 | 60.51 |
| *aac(6')-Ib* | 1 | 100 | 0.99 | 12 | 44 | 21.43 | 0.000017 | 13 | 144 | 8.28 |

Note: -, indicates no results available.

P-value: *P* < 0.05 was considered statistically significant between Dujiang Yan and Barkam.

### 3.3 Associations for 21 AMRs/11 ARGs, between AMRs and ARGs, respectively

For the associations between AMRs pairs, 58 pairs were found positively associated (OR > 1) (Table 5). The strong association was observed between CXM and AMP (OR, 828.000; 95% CI, 93.653–7320.478), followed by KZ/CTX (OR, 497.250; 95% CI, 87.295–2832.443), CXM/ CTX (OR, 328.667; 95% CI, 70.110–1540.743), ATM/CRO (OR, 223.667; 95% CI, 27.667–1808.144) and ATM/KZ (OR, 188.571; 95% CI, 23.581–1507.989). A total of 16 pairs positive associations (OR > 1) in ARGs were also observed (Table 6). The strongest positive association was detected between *sul1* and *floR* (OR, 21.667; 95% CI, 6.361–73.802), followed by *sul1/sul2* (OR, 17.143; 95% CI, 6.026–48.766) and *bla*$_{CTX-M}$/*sul3* (OR, 15.278; 95% CI, 4.937–47.282).

As Table 7 showed, a total of 73 pairs positive associations between AMRs and ARGs were observed (*P* < 0.05) and no negative associations was observed. Positive associations were found between 9 pairs of AMR phenotypes with corresponding resistance genes. There was a positive association between chloramphenicol antibiotics with *floR* gene, CHL/*floR* (OR: 301.167, CI: 46.078–1968.425). Six β-lactams antibiotics were positively associated with *bla*$_{CTX-M}$ gene: CRO/*bla*$_{CTX-M}$ (OR: 27.857, CI: 9.579–81.011), KZ/*bla*$_{CTX-M}$ (OR: 23.958, CI: 8.371–68.573), CXM/*bla*$_{CTX-M}$ (OR: 22.353, CI: 7.862–63.557), CTX/*bla*$_{CTX-M}$ (OR: 20.926, CI: 7.403–59.151), ATM/*bla*$_{CTX-M}$ (OR: 15.815, CI: 5.668–44.125) and CAZ/*bla*$_{CTX-M}$ (OR: 6.526, CI: 2.059–20.685). Two tetracyclines antibiotics have positive associations with *tetB* gene: DO/ *tetB* (OR: 15.030, CI: 3.698–61.087) and TE/*tetB* (OR: 9.892, CI: 2.477–39.515).

### 3.4 Integron types and gene cassette

Among the 157 strains, *int1* and *intI2* genes were found in 16 (10.19%, 16/157) and 2 (1.27%, 2/157) strains, respectively (Table 8). As Table 8 showed, only 1 strain (Bar 36) contained both

**Table 4. Distribution of antibiotic resistance phenotypes and antimicrobial resistance genes detected in *E. coli* strains isolated from musk deer (n = 157).**

| Class of antibiotic | Antimicrobial agents | Resistant rate (%) | Intermediate rate (%) | Sensitive rate (%) | Resistance genes | Positive rate (%) |
|---|---|---|---|---|---|---|
| Aminoglycosides | Gentamicin | 6.37 (10/157) | 1.91 (3/157) | 91.72 (144/157) | $aac(3')$-$Iid$ | 60.51 (10/157) |
| | Tobramycin | 3.18 (5/157) | 3.18 (5/157) | 93.63 (147/157) | | |
| | Amikacin | 0.00 (0/157) | 0.64 (1/157) | 99.36 (156/157) | $aac(6')$-$Ib$ | 8.28 (13/157) |
| | Netilmicin | 1.27 (2/157) | 2.55 (4/157) | 96.18 (151/157) | | |
| Chloram phenicols | Chloramphenicol | 10.19 (16/157) | 3.82 (6/157) | 85.99 (135/157) | $catA1$ | 0 (0/157) |
| | | | | | $flor$ | 9.55 (15/157) |
| Quinolones | Ciprofloxacin | 15.92 (25/157) | 4.46 (7/157) | 79.62 (125/157) | $qnrA$ | 1.27 (2/157) |
| Sulfonamide | Sulfamethoxazole | 85.99 (135/157) | 3.82 (6/157) | 10.83 (16/157) | $sul1$ | 14.01 (22/157) |
| | | | | | $sul2$ | 19.11 (30/157) |
| | | | | | $sul3$ | 10.83 (17/157) |
| Tetracyclines | Tetracycline | 24.84 (39/157) | 5.10 (8/157) | 70.06 (110/157) | $tetA$ | 12.10 (19/157) |
| | Doxycycline | 19.11 (30/157) | 8.92 (14/157) | 71.97 (113/157) | | |
| | Tigecycline | 0.64 (1/157) | 19.11 (30/157) | 80.25 (126/157) | $tetB$ | 7.01 (11/157) |
| | Minocycline | 5.73 (9/157) | 12.74 (20/157) | 81.53 (128/157) | | |
| β-lactams | Piperacillin/tazobactam | 1.91 (3/157) | 1.91 (3/157) | 96.18 (151/157) | $bla_{TEM}$ | 91.72 (144/157) |
| | Ertapenem | 0.00 (0/157) | 0.00 (0/157) | 100.00 (157/157) | | |
| | Imipenem | 0.00 (0/157) | 1.91 (3/157) | 98.09 (154/157) | | |
| | Meropenem | 0.00 (0/157) | 0.00 (0/157) | 100.00 (157/157) | | |
| | Cephazolin | 22.93 (57/157) | 0.00 (0/157) | 77.07 (100/157) | | |
| | Cefuroxime sodium | 23.57 (37/157) | 40.76 (64/157) | 35.67 (56/157) | | |
| | Cefotaxime | 24.20 (38/157) | 21.02 (33/157) | 54.78 (86/157) | $bla_{CTX-M}$ | 15.56 (26/157) |
| | Ceftriaxone | 21.66 (34/157) | 0.64 (1/157) | 77.71 (122/157) | | |
| | Ceftazidime | 8.92 (14/157) | 8.92 (14/157) | 82.17 (129/157) | | |
| | Cefepime | 9.55 (15/157) | 10.19 (16/157) | 80.25 (126/157) | | |
| | Cefoxitin | 0.00 (0/157) | 0.00 (0/157) | 100.00 (157/157) | | |
| | Aztreonam | 14.65 (23/157) | 3.18 (5/157) | 82.17 (129/157) | $bla_{SHV}$ | 0.00 (0/157) |
| | Ampicillin | 26.11 (41/157) | 15.92 (25/157) | 57.96 (91/157) | | |
| | Amoxicillin/clavulanic acid | 0.64 (1/157) | 6.37 (10/157) | 92.99 (146/157) | | |
| | Ampicillin/sulbactam | 1.91 (3/157) | 1.91 (3/157) | 96.18 (151/157) | | |
| | Fosfomycin | 1.27 (2/157) | 0.00 (0/157) | 98.73 (155/157) | | |

*intI1* and *intI2* genes. No *intI3* gene was detected. The prevalence of class 1 integrons was 8.91% (9/101) in Dujiang Yan and 12.50% (7/56) in Barkam. Class 2 integrons was only detected in Barkam (3.57%, 2/56). As shown in Table 8, only one cassette arrays (*dfrA17-aadA5*) were identified from 16 class 1 integron positive isolates, no cassette was detected in class 2 integron.

## 4. Discussion

To better understand the antibiotic resistance of *E. coli* from captive musk deer in Sichuan, China, 157 *E. coli* strains isolated were detected for the occurrence of AMRs, ARGs, integrons and gene cassettes. Among the 157 *E. coli* strains, 144 strains were resistant to at least one antibiotic (91.72%, 144/157), which is lower than data from hybrid deer as reported in another study (100.00%, 51/51) [23]. Meanwhile, 38 isolates in our study were found to be multi-drug resistant (MDR) and the rate of MDR (24.2%, 38/144) is higher than *E. coli* strains isolated from hybrid deer (3.30%, 1/30) [24]. However, the rate of MDR in our result is lower than the rate in captive giant pandas (100.00%, 89/89) and non-human primates (49.65%, 494/995) [16,

**Table 5. The associations between AMRs among *E. coli* isolates from musk deer (n = 157).**

| AMRs | RL (135) | AMP (41) | TE (39) | CTX (38) | CXM (37) | KZ (36) | CRO (34) | DO (30) | CIP (25) | ATM (23) | C (16) | CAZ (14) | CN (10) | MH (18) | TOB (5) | TZP (3) | SAM (3) | NET (18) | FOS (18) | TGC (18) | AMC (18) |
|---|---|---|---|---|---|---|---|---|---|---|---|---|---|---|---|---|---|---|---|---|---|
| RL (135) | - | NS | - | - | - | - | - | - | - | - | NA | - | NA | - | NA | NA | - | NA | NA | NA | NA |
| AMP (41) | NS | - | 2.212 (1.016–4.816) | 136.000 (37.548–492.590) | 828.000 (93.653–7320.478) | NA | NA | 2.704 (1.173–6.233) | 15.833 (5.677–44.158) | 133.158 (16.938–1046.797) | 29.556 (6.337–137.842) | 53.393 (6.701–425.448) | NA | - | NA | NA | NA | NA | NA | NA | NA |
| TE (39) | - | 2.212 (1.016–4.816) | - | NS | - | - | - | 86.250 (22.748–327.021) | - | - | 4.757 (1.637–13.826) | - | 5.182 (1.380–19.459) | 6.970 (1.653–29.386) | - | - | NA | NA | NA | NA | NA |
| CTX (38) | - | 136.000 (37.548–492.590) | NS | - | 328.667 (70.110–1540.743) | 497.250 (87.295–2832.443) | NA | - | 11.232 (4.296–29.367) | 162.250 (20.455–1286.988) | 13.269 (3.961–44.453) | 61.360 (7.672–490.773) | 36.621 (4.460–300.713) | - | NA | NA | NA | NA | NA | NA | NA |
| CXM (37) | - | 828.000 (93.653–7320.478) | - | 328.667 (70.110–1540.743) | - | NS | NA | - | 20.056 (7.062–56.958) | 174.533 (21.920–1389.685) | 81.136 (10.190–646.031) | NA | 38.250 (4.653–314.424) | 4.531 (1.149–17.864) | NA | NA | NA | NA | NA | NA | NA |
| KZ (36) | - | NA | - | 497.250 (87.295–2832.443) | NS | - | NS | - | 16.286 (5.963–44.475) | 188.571 (23.581–1507.989) | 37.864 (8.038–178.365) | 67.826 (8.454–544.164) | 40.000 (4.860–329.185) | - | NA | NA | NA | NA | NA | NA | NA |
| CRO (34) | - | NA | - | NA | NA | NS | - | - | 11.259 (4.328–29.290) | 223.667 (27.667–1808.144) | 16.227 (4.793–54.942) | 75.524 (9.379–608.162) | 43.920 (5.323–362.377) | - | NA | NA | - | NA | NA | NA | NA |
| DO (30) | - | 2.704 (1.173–6.233) | 86.250 (22.748–327.021) | - | - | - | - | - | 2.973 (1.161–7.615) | - | 5.409 (1.836–15.934) | - | 4.880 (1.314–18.125) | 45.818 (5.458–384.635) | NA | - | - | NA | - | NA | NA |
| CIP (25) | - | 15.833 (5.677–44.158) | - | 11.232 (4.296–29.367) | 20.056 (7.062–56.958) | 16.286 (5.963–44.475) | 11.259 (4.328–29.290) | 2.973 (1.161–7.615) | - | 13.217 (4.787–36.488) | 10.045 (3.289–30.676) | 33.786 (8.410–135.731) | 10.105 (2.610–39.129) | 8.000 (1.979–32.334) | 8.864 (1.400–56.111) | NA | NA | - | - | NA | NA |
| ATM (23) | - | 133.158 (16.938–1046.797) | - | 162.250 (20.455–1286.988) | 174.533 (21.920–1389.685) | 188.571 (23.581–1507.989) | 223.667 (27.667–1808.144) | - | 13.217 (4.787–36.488) | - | 11.663 (3.762–36.159) | 172.900 (20.484–1459.403) | 11.471 (2.937–44.799) | - | 9.900 (1.557–62.960) | NA | - | - | NA | NA | NA |
| CHL (16) | NA | 29.556 (6.337–137.842) | 4.757 (1.637–13.826) | 13.269 (3.961–44.453) | 81.136 (10.190–646.031) | 37.864 (8.038–178.365) | 16.227 (4.793–54.942) | 5.409 (1.836–15.934) | 10.045 (3.289–30.676) | 11.663 (3.762–36.159) | NS | 14.889 (4.282–51.771) | 12.364 (3.100–49.316) | 9.067 (2.146–38.313) | 16.038 (2.454–104.827) | 20.000 (1.704–234.687) | - | NA | - | NA | NA |
| CAZ (14) | - | 53.393 (6.701–425.448) | - | 61.360 (7.672–490.773) | NA | 67.826 (8.454–544.164) | 75.524 (9.379–608.162) | - | 33.786 (8.410–135.731) | 172.900 (20.484–1459.403) | 14.889 (4.282–51.771) | - | 15.333 (3.739–62.875) | - | 19.227 (2.900–127.460) | NA | 23.667 (1.998–280.277) | - | NA | NA | NA |
| CN (10) | NA | NA | 5.182 (1.380–19.459) | 36.621 (4.460–300.713) | 38.250 (4.653–314.424) | 40.000 (4.860–329.185) | 43.920 (5.323–362.377) | 4.880 (1.314–18.125) | 10.105 (2.610–39.129) | 11.471 (2.937–44.799) | 12.364 (3.100–49.316) | 15.333 (3.739–62.875) | - | NS | 97.333 (9.390–1008.882) | 36.500 (2.985–446.324) | NS | NS | NA | NA | NA |
| MH (18) | - | - | 6.970 (1.653–29.386) | - | 4.531 (1.149–17.864) | - | - | 45.818 (5.458–384.635) | 8.000 (1.979–32.334) | - | 9.067 (2.146–38.313) | - | NS | - | NS | NA | - | NA | NA | NA | NA |
| TOB (5) | NA | NA | - | NA | NA | NA | NA | NA | 8.864 (1.400–56.111) | 9.900 (1.557–62.960) | 16.038 (2.454–104.827) | 19.227 (2.900–127.460) | 97.333 (9.390–1008.882) | NS | - | - | - | NS | NA | NA | NA |
| TZP (3) | NA | NA | - | NA | NA | NA | NA | - | NA | NA | 20.000 (1.704–234.687) | NA | 36.500 (2.985–446.324) | NA | - | - | - | - | - | NA | NA |
| SAM (3) | - | NA | NA | NA | NA | NA | - | - | NA | - | - | 23.667 (1.998–280.277) | NS | - | - | - | - | NS | NA | NA | NA |
| NET (18) | NA | NA | NA | NA | NA | NA | NA | NA | - | - | NA | - | NS | NA | NS | - | NS | - | NA | NA | NA |
| FOS (18) | NA | NA | NA | NA | NA | NA | NA | - | - | NA | - | NA | NA | NA | NA | - | NA | NA | - | NS | NA |
| TGC (18) | NA | NA | NA | NA | NA | NA | NA | NA | NA | NA | NA | NA | NA | NA | NA | NA | NA | NA | NS | - | NA |
| AMC (18) | NA | NA | NA | NA | NA | NA | NA | NA | NA | NA | NA | NA | NA | NA | NA | NA | NA | NA | NA | NA | NS |

Note: Only AMRs with a significant association (*P* < 0.05) are shown.

Odds ratio (OR) for significant associations between AMRs (95% confidence interval in parenthesis); Na, no results available (or) could not be calculated because none of the isolates carried one of the combinations of AMR or one of the values of antibiotic was a constant or zero);—indicates no significant associations (*P* > 0.05); NS, no statistics were determined for the same AMR.

**Table 6. The associations between ARGs among *E. coli* isolates from musk deer (n = 157).**

| ARGs | *bla$_{TEM}$* (144) | *aac(3')-Iid* (95) | *sul2* (30) | *bla$_{CTX-M}$* (26) | *sul1* (22) | *tetA* (18) | *sul3* (17) | *flor* (15) | *aac(6')-Ib* (13) | *tetB* (11) | *qnrA* (18) |
|---|---|---|---|---|---|---|---|---|---|---|---|
| *bla$_{TEM}$*(144) | NS | - | - | NA | - | NA | NA | - | - | NA | NA |
| *aac(3')-Iid* (95) | - | NS | - | 3.235(1.150–9.104) | 4.917(1.389–17.407) | - | 12.354(1.594–95.749) | - | 8.819(1.117–69.654) | - | NA |
| *sul2*(30) | - | - | NS | 3.469(1.379–8.725) | 17.143(6.026–48.766) | 5.014(1.820–13.812) | 3.561(1.228–10.322) | NA | 8.873(2.656–29.637) | - | NA |
| *bla$_{CTX-M}$*(26) | NA | 3.235(1.150–9.104) | 3.469(1.379–8.725) | NS | - | - | 15.278(4.937–47.282) | - | - | - | - |
| *sul1*(22) | - | 4.917(1.389–17.407) | 17.143(6.026–48.766) | - | NS | 4.783(1.632–14.019) | - | 21.667(6.361–73.802) | 6.857(2.049–22.948) | - | NA |
| *tetA*(18) | NA | - | 5.014(1.820–13.812) | - | 4.783(1.632–14.019) | NS | - | 13.610(4.156–44.577) | - | - | NA |
| *sul3*(17) | NA | 12.354(1.59495.749) | 3.561(1.22810.322) | 15.278(4.937–47.282) | - | - | NS | 5.417(1.590–18.450) | - | - | - |
| *flor*(15) | - | - | NA | - | 21.667(6.361–73.802) | 13.610(4.156–44.577) | 5.417(1.590–18.450) | NS | 5.374(1.423–20291) | - | NA |
| *aac(6')-Ib* (13) | - | 8.819(1.11769.654) | 8.873(2.65629.637) | - | 6.857(2.049–22.948) | - | - | 5.374(1.423–20291) | NS | - | NA |
| *tetB*(11) | NA | - | - | - | - | - | - | - | - | NS | NA |
| *qnrA*(18) | NA | NA | NA | - | NA | NA | - | NA | NA | NA | NS |

Note: Only ARGs with a significant association ($P < 0.05$) are shown.

Odds ratio (OR) for significant associations between resistance genes (95% confidence interval in parenthesis); Na, no results available (or) could not be calculated because none of the isolates carried one of the combinations of resistance genes or one of the value of genes was a constant or zero);—indicates no significant associations ($P > 0.05$); NS, no statistics were determined for the same gene.

25], which showed MDR strains from musk deer is not serious compared with giant pandas and non-human primates. Our results showed *E. coli* from musk deer were mostly resistant to RL (85.99%), AMP (26.11%) and TE (24.84%) and the antibiotic resistance rates are lower than the corresponding resistance rate of *E. coli* from deer (RL, AMP, TE were 90.63%, 90.63% and 87.50%, respectively) reported in Bangladesh Nature Reserve [26]. This phenomenon may relate to different antibiotic used in the two regions [27, 28]. However, the resistance rate against TE (24.84%), KZ (22.93%), ATM (14.65%), CHL (10.19%), CAZ (8.92%) and CN (6.37%) were higher than the rate previously reported from captive musk deer in Sichuan, China (TE, 20.34%; KZ, 8.47%, ATM, 1.69%; CHL, 6.78%; CAZ, 1.69% and CN, 5.08%) [29]. This implied the antimicrobial resistance rate of *E. coli* in musk deer has increased [30, 31]. It is worth noting that our results showed that 85.53% of *E. coli* are resistant to RL, but two farms have not used any sulfonamides in recent years. This also suggests that there are more complex mechanisms for the generation of bacterial drug resistance in addition to antibiotic pressure [32, 33]. The rate of MDR from Barkam is higher than that from Dujiang Yan, indicated antibiotic resistant in Barkam is more severe and more caution should be exercised when choosing

**Table 7. The associations between AMRs and ARGs among *E. coli* isolates from musk deer (n = 157).**

| AMR/ARGs | *bla*$_{TEM}$ (144) | *aac(3')-IId* (95) | *sul2* (30) | *bla*$_{CTX-M}$ (26) | *sul1* (22) | *tetA* (18) | *sul3* (17) | *flor* (15) | *aac(6')-Ib* (13) | *tetB* (11) | *qnrA* (18) |
|---|---|---|---|---|---|---|---|---|---|---|---|
| RL (135) | - | - | - | - | - | - | - | - | - | - | NA |
| AMP (41) | - | - | 4.792 (1.827–12.567) | - | 8.542 (3.057–23.864) | - | 4.227 (1.376–12.991) | 7.408 (2.353–23.324) | - | 4.063 (1.081–15.270) | NA |
| TE (39) | - | - | 3.570 (1.541–8.270) | - | 4.800 (1.876–12.278) | - | - | 4.092 (1.376–12.168) | - | 9.892 (2.477–39.515) | NA |
| CTX (38) | - | - | 5.455 (2.327–12.786) | 20.926 (7.403–59.151) | 2.531 (0.985–6.504) | - | 5.714 (1.998–16.344) | 11.713 (3.462–39.623) | - | - | - |
| CXM (37) | - | - | 10.460 (4.276–25.584) | 22.353 (7.862–63.557) | 5.280 (2.052–13.583) | - | 8.038 (2.724–23.720) | 72.435 (9.074–578.235) | 4.433 (1.387–14.174) | - | NA |
| KZ (36) | - | - | 33.630 (7.108–159.115) | 23.958 (8.371–68.573) | 4.400 (1.716–11.284) | - | 6.264 (2.180–18.001) | 33.630 (7.108–159.115) | 3.257 (1.019–10.413) | - | NA |
| CRO (34) | - | - | 6.921 (2.889–16.578) | 27.857 (9.579–81.011) | 3.046 (1.173–7.911) | - | 6.905 (2.389–19.953) | 14.228 (4.166–48.599) | - | - | - |
| DO (30) | - | - | 4.948 (2.043–11.983) | - | 10.026 (3.724–26.993) | - | - | 4.527 (1.495–13.713) | 4.286 (1.323–13.881) | 15.030 (3.698–61.087) | NA |
| CIP (25) | - | 5.927 (1.691–20.773) | 14.984 (5.585–40.199) | 6.129 (2.358–15.930) | 8.643 (3.172–23.548) | - | 11.905 (3.945–35.922) | 11.813 (3.716–37.552) | 18.000 (4.968–65.218) | - | - |
| ATM (23) | NA | - | 5.548 (2.143–14.365) | 15.815 (5.668–44.125) | - | - | 3.947 (1.292–12.054) | 4.902 (1.551–15.490) | 1.860 (0.471–7.349) | - | NA |
| CHL (16) | - | - | 126.000 (15.525–1022.613) | - | 17.917 (5.548–57.860) | 8.361 (2.644–26.441) | - | 301.167 (46.078–1968.425) | - | - | - |
| CAZ (14) | NA | - | 15.375 (4.396–53.778) | 6.526 (2.059–20.685) | - | - | 4.000 (1.099–14.562) | 7.389 (2.079–26.262) | 5.956 (1.557–22.782) | - | NA |
| CN (10) | NA | - | 12.580 (3.029–52.243) | - | 38.000 (7.339–196.768) | - | - | 8.242 (2.020–33.641) | - | - | NA |
| MH (18) | - | - | 19.022 (3.715–97.393) | - | 9.632 (2.355–39.396) | - | 8.308 (1.983–34.809) | 17.250 (3.994–74.508) | 12.356 (2.820–54.144) | - | - |
| TOB (5) | NA | - | NA | - | 10.500 (1.647–66.955) | NA | NA | 17.500 (2.660–115.125) | - | NA | NA |
| TZP (3) | NA | NA | NA | - | - | NA | - | - | - | NA | NA |
| SAM (3) | - | NA | 14.000 (1.207–162.356) | - | - | - | NA | - | NA | NA | NA |
| NET (18) | NA | - | NA | NA | NA | NA | NA | NA | NA | NA | NA |
| FOS (18) | NA | - | - | - | - | - | NA | - | NA | - | NA |
| TGC (18) | NA | NA | NA | NA | NA | NA | NA | NA | NA | NA | NA |

(*Continued*)

**Table 7.** (Continued)

| AMR/ ARGs | $bla_{TEM}$ (144) | aac(3')- IId (95) | sul2 (30) | $bla_{CTX-M}$ (26) | sul1 (22) | tetA (18) | sul3 (17) | flor (15) | aac(6')- Ib (13) | tetB (11) | qnrA (18) |
|---|---|---|---|---|---|---|---|---|---|---|---|
| AMC (18) | NA | NA | NA | NA | NA | NA | NA | NA | NA | NA | NA |

ote: Only AMRs and ARGs with a significant association ($P < 0.05$) are shown.

Odds ratio (OR) for significant associations between AMRs and ARGs (95% confidence interval in parenthesis); Na, no results available (or) could not be calculated because none of the isolates carried one of AMR or one of ARGs was a constant or zero);—indicates no significant associations ($P > 0.05$).

antibiotics. The antibiotic resistance rates of Barkam to TE, KZ, CXM, SAM and CIP are significantly higher than that from Dujiang Yan, which may relate to different living environments [34, 35].

The expression of antibiotic resistance genes is an important mechanism of antimicrobial resistance [36]. In our study, 11 ARGs were detected and the most prevalent ARG was $bla_{TEM}$ (91.72%), the occurrence of which is higher than that from another study on musk deer (30.51%) [29]. However, the detection rate of another ESBL-encoding gene $bla_{CTX-M}$ (16.56%) in our present study is lower than the previous result (35.59%) [29]. As important ESBL-encoding genes, $bla_{TEM}$ and $bla_{CTX-M}$ were both detected in *E. coli* implied feces excretion from musk deer posed a potential harm to environmental safety [37]. The prevalence of *tetB* (7.01%) and *sul3* (10.83%) from musk deer in our study were higher than the prevalence previously reported from other deer in Northeastern China [38]. Moreover, *aac(3')-IId* (60.51%), *aac(6')-Ib* (8.28%) and *qnrA* (1.27%) genes were firstly detected in *E. coli* from musk deer. The above results suggest the carried rate of ARGs in musk deer are increasing. ARGs carried in clinically healthy animal (non-human primates, giant pandas and waterfowl birds) pose a potential threat to the surrounding environment [16, 39, 40]. Our results showed that musk deer carried 11 kinds of ARGs and the surrounding environment was a potential threat by feces of musk deer, suggesting that the feces should be disinfected to reduce the environmental pollution caused by antibiotic-resistant *E. coli*. High prevalence of ARGs implied that *E. coli* from musk deer is a reservoir of ARGs, and continuously monitoring the occurrence rate of ARGs in *E. coli* from musk deer is needed in future.

Previous studies have shown that an association exist between AMR and ARGs in *E. coli* [41–43]. Our result showed only positive associations (OR > 1) were observed between ARGs and AMRs, no negative associations were observed. This is similar to the another report, where only positive associations between AMRs and ARGs in Urinary tract patient were detected [44]. Our study showed the AMR-ARG pair with the strongest positive association was CHL/*floR* (OR, 301.167; 95% CI, 46.078–1968.425). This is consistent with the results of *E. coli* from pigs, since chloramphenicol is positively associated with the corresponding AGRs [45]. Although 9 pairs of AMRs and ARGs showed positive associations in our result, the antibiotic resistance

**Table 8. Integrons and gene cassettes detected among *E. coli* isolates from musk deer.**

| Strains number | Types of Integrons | Gene cassettes |
|---|---|---|
| DJY6, DJY23, DJY36, DJY42, DJY82, Bar2, Bar11, Bar16, Bar33, Bar35, Bar43 | class1 | - |
| DJY13, DJY53, DJY75, DJY81, Bar1 | class1 | *dfrA17-aadA5* |
| Bar33, Bar48 | class2 | - |

Note: -, No gene cassette was detected

phenotypes did not completely match the ARGs, which is similar to the findings from previous studies [46–48]. In general, the antibiotic resistant phenotype of bacteria is determined by its ARGs [49–51]. In our study, antibiotic resistant phenotype was not completely consistent to the ARGs detected. The reason may relate to the expression level of ARGs, which may not be high enough to exert drug resistance [52]. Moreover, our present study using PCR to detect ARGs, not using whole gene sequencing, which may underestimate the detection rate of ARGs.

Previous studies showed integron plays a vital role in the dissemination of ARGs [30, 53, 54]. In our results, class 1 and class 2 integrons were all detected from musk deer, while class 3 integron was not detected. Class 1 integron (10.19%) was more prevalent compared with class 2 integron (1.27%), which is similar to previous reports [55–57]. Gene cassette that harbored in integrons-positive isolates is an important media to spread ARGs to the environment [58, 59]. The distribution rate of integron (10.82%) from musk deer in our study is lower than that in other captive animals, such as little penguins (44.70%), Australian sea lion (50.00%) and giant panda (35.71%) [56, 60–62]. Our result showed only *dfrA17-aadA5* gene cassettes arrays were detected in class 1 integron-positive isolates. Trimethoprim resistance gene (*dfrA17*) and aminoglycoside resistance gene (*aadA5*) was observed for the first time in musk deer in our present study. The type of integron gene cassettes carried by *E. coli* from forest musk deer are lower than that from other animals, such as giant panda (6 types), yak (11 types) and non-human primates (18 types) [16, 25, 63]. The results showed that there were a few kinds of integrons and gene cassettes carried by intestinal *E. coli* from forest musk deer. Positive results of class 1 and class 2 integrons detected in *E. coli* from forest musk deer implied drug resistance and horizontal transfer of ARGs cannot be ignored.

## 5. Conclusion

This is the first study detected the prevalence of antimicrobial resistance, antibiotic resistance genes and integron genes cassettes in *E. coli* obtained from fecal of musk deer. Positive associations were more common observed among *E. coli* strains carrying antibiotic resistance genes and antimicrobial resistance. Our results implied musk deer might act as an important reservoir of ARGs and integrons, which may spread into the natural environment and accompanying animals around musk deer.

## Supporting information

**S1 Table. Resistance profile and integron from musk deer.**
(XLS)

**S2 Table. Primers used in the study.**
(XLS)

## Acknowledgments

We thank Dr Junai Gan for the English language revision.

## Author Contributions

**Conceptualization:** Liqin Wang, Guangneng Peng, Zhijun Zhong.

**Data curation:** Hang Liu, Shulei Pan, Lijun Luo, Siping Fan, Shaoqi Jiang, Haifeng Liu, Zhijun Zhong.

**Investigation:** Yuehong Cheng, Ziyao Zhou, Zhihua Ren, Xiaoping Ma, Dongjie Cai.

**Methodology:** Liqin Wang, Guangneng Peng, Zhijun Zhong.

**Software:** Liqin Wang, Suizhong Cao, Ya Wang, Liping Gou, Yi Geng, Guangneng Peng, Qigui Yan, Zhijun Zhong.

**Supervision:** Lei Zhou.

**Validation:** Liuhong Shen, Ya Wang, Liping Gou, Yi Geng, Qigui Yan.

**Visualization:** Yuehong Cheng, Ziyao Zhou, Zhihua Ren, Xiaoping Ma, Dongjie Cai.

**Writing – original draft:** Hang Liu, Shulei Pan, Lijun Luo, Siping Fan, Shaoqi Jiang, Haifeng Liu, Zhijun Zhong.

**Writing – review & editing:** Shaqiu Zhang, Yan Luo, Zhijun Zhong.

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
