## [Decision Letter · Decision Letter 0]

6 Jun 2023

PONE-D-23-16177Distribution and Associations for Antimicrobial Resistance and Antibiotic Resistance Genes of Escherichia coli from musk deer (Moschus berezovskii) in Sichuan, ChinaPLOS ONE

Dear Dr. Zhong,

Thank you for submitting your manuscript to PLOS ONE. After careful consideration, we feel that it has merit but does not fully meet PLOS ONE’s publication criteria as it currently stands. Therefore, we invite you to submit a revised version of the manuscript that addresses the points raised during the review process.

We look forward to receiving your revised manuscript.

Kind regards,

Professor Md. Tanvir Rahman, DVM, MSc, PhD

Academic Editor

PLOS ONE

Journal Requirements:

   "This work was funded by the National Key Research and Development Program of China (2018YFD0500900, 2016YFD0501009), the Chengdu Giant Panda Breeding 

Research Foundation (CPF2017-05, CPF2015-4) and the Science and Technology Achievements Transfer Project in Sichuan province (2022JDZH0026). We thank Dr 

Junai Gan for the English language revision."

   "The author(s) received no specific funding for this work"

5. Please include your tables as part of your main manuscript and remove the individual files. Please note that supplementary tables (should remain/ be uploaded) as separate "supporting information" files.

Additional Editor Comments:

Dear authors,

its a nice study. Please see the comments of the reviewers and take te necessary actions as suggested to revise the manuscriot.

Best wishes,

Tanvir

==

Md. Tanvir Rahman, DVM, MSc. (Canada), Ph.D. (UK), Postdoc (Germany)

ACADEMIC EDITOR , PLOS ONE

&

Professor,

Department of Microbiology and Hygiene,

Faculty of Veterinary Science,

Bangladesh Agricultural University, Mymensingh-2202, Bangladesh.

Mobile. +88-01913323307

Fax.+88-02996661510

E.mail: tanvirahman@bau.edu.bd

http://vmh.bau.edu.bd/profile/VMH1005

https://orcid.org/0000-0001-5432-480X

https://sites.google.com/site/tanvirahman/Home

https://www.researchgate.net/profile/Mdtanvir-Rahman

Reviewers' comments:

Reviewer's Responses to Questions

**Comments to the Author**

1. Is the manuscript technically sound, and do the data support the conclusions?

Reviewer #1: Yes

Reviewer #2: Yes

2. Has the statistical analysis been performed appropriately and rigorously? 

Reviewer #1: Yes

Reviewer #2: Yes

3. Have the authors made all data underlying the findings in their manuscript fully available?

Reviewer #1: Yes

Reviewer #2: Yes

4. Is the manuscript presented in an intelligible fashion and written in standard English?

Reviewer #1: Yes

Reviewer #2: Yes

5. Review Comments to the Author

Reviewer #1: This is an interesting piece of work about the antimicrobial resistance of E. coli isolates from musk deer. The study aimed to characterize the antimicrobial resistance of 157 E. coli isolates from the feces of captive musk deer, especially antibiotic resistance genes and integron gene cassettes. The authors found high rates of multi-drug resistances and high diversity of antibiotic resistance genes detected. Meanwhile, the correlation between antibiotics resistance phenotype and antibiotics resistance genes was also analyzed. This is the first study detected the prevalence of antimicrobial resistance, antibiotic resistance genes and integron genes cassettes in E. coli obtained from musk deer.

I have some comments that would help clarify the presentation of this study.

Question 1: The authors mention that antibiotic-resistant E. coli carrying ARGs in musk deer’s feces that are discharged into the environment may become pollutants for nature. How about some specific suggestions on the implementation of solve this problem?

Question 2: Were the isolates also tested for colistin/polymyxin? In the current scenario, it would be interesting to search for the mcr genes in this study.

Question 3: Were PCR positive and negative controls performed in their study?

Question 4: In Table 1 and Table 4, unit % is given at the top, so there is no need to duplicate it in the content.

Question 5: Line 68-69, Please list detailed species that have detected AMR associated with ARGs.

Question 6: Please analyze the difference of drug resistance between Dujiang Yan and Barkam in the discussion part.

Reviewer #2: This manuscript aimed to investigate the antimicrobial resistance (AMR), antibiotic resistance genes (ARGs), and integrons in 157 Escherichia coli strains isolated from feces of captive musk deer in Sichuan province. The results implied musk deer is a reservoir of ARGs and positive associations were common observed among E. coli strains carrying AMRs and ARGs. It's provided scientific value to this field. But, I have some questions as follows.

1. In Sample collection, how to know the feces samples from each animal accurately? Please explain methods of collection samples carefully, and to ensure each samples correspondence to each animal.

2. The results found that there has some difference in AMR/MDR between Dujiang Yan and Barkam, and why this phenomenon was not mentioned in the discussion?

3. In this study, Escherichia coli strains from clinically healthy captive musk deer carrying a large number of ARGs, which make a little difficult to understanding this phenomenon. I suggest in the discussion the author should consider add some information to explain this phenomenon.

4. Table 8 may not be well expressed, which need revised concise to better understanding. Meanwhile, only one cassette arrays (dfrA17-aadA5) was identified. Have the results been repeated three times to make sure the accuracy of their results?

6. PLOS authors have the option to publish the peer review history of their article (what does this mean?). If published, this will include your full peer review and any attached files.

Reviewer #1: No

Reviewer #2: No

---

## [Author Response · Author response to Decision Letter 0]

21 Jun 2023

Dear Editor:

 Thank you very much for your letter and the reviewers’ comments regarding our manuscript submitted to “PLOS ONE” (Manuscript ID: PONE-D-23-16177). We have checked the article and revised it according to the comments, and carefully proof-read the manuscript. All revisions were marked in revised manuscript with track changes. 

 We hope, with these modifications and improvements based on your suggestion and the reviewer’s comments, the quality of our manuscript would meet the publication standard of “PLOS ONE”. Once again, we acknowledge your comments and constructive suggestions very much, which are valuable in improving the quality of our manuscript.

 We submit here the revised manuscript with track changes as well as a list of changes. Thanks again for your reconsideration of our manuscript for publication in your journal. If you have any question about this paper, please don’t hesitate to let me know.

 Best wishes for you!

Sincerely yours,

Zhijun Zhong 

College of Veterinary Medicine

Key Laboratory of Animal Disease and Human Health of Sichuan Province

Sichuan Agricultural University

Chengdu 611130, P. R. China

E-mail: zhongzhijun488@126.com

Response to Reviewer 1 Comments

Response to Reviewer:

 Thank you very much for your time and thoughtful comments, many of which have been incorporated into the revised manuscript (Tracked Version). Detailed responses are as follows.

Question 1: The authors mention that antibiotic-resistant E. coli carrying ARGs in musk deer’s feces that are discharged into the environment may become pollutants for nature. How about some specific suggestions on the implementation of solve this problem?

Response 1: Thanks for your comments. We add some suggestions in our revised manuscript (Lines 249-251): We suggest that the feces forest musk deer should be disinfected to reduce the environmental pollution caused by antibiotic-resistant E. coli.

Question 2: Were the isolates also tested for colistin/polymyxin? In the current scenario, it would be interesting to search for the mcr genes in this study.

Response 2: Thanks for your comments. AMC, CN, CRO, AK, ENR, FFC and FOX have been used in Sichuan Institute of Musk Deer Breeding for disease treatment and control. Colistin/polymyxin was not used in the two farms. As the reviewer suggestion, colistin/polymyxin resistance is a topic for scientists in recent years. In our ongoing research, we conducted a preliminary experiment to detect mcr gene. However, colistin resistance gene mcr was not detected in all E. coli isolated from musk deer (data not showed in our current manuscript). Moreover, mcr gene was also not detected in E. coli isolated from other captive wild animals (monkey, black bear, golden pheasant, and silver pheasant) in our laboratory (data not published).

Question 3: Were PCR positive and negative controls performed in their study?

Response 3: Thanks for your comments. We have set up a negative control, but antibiotic resistance genes (ARGs) amplify do not need positive control in general. There have many researches amplify ARGs without positive control (1, 2). In our study, we used primers derived from references and the length of the PCR products was determined and compared to the 2000 DNA Marker (Sangon Biotech, Shanghai, China) during gel electrophoresis. Notably, for the confirmation of each ARG, the PCR products were purified and sequenced by Sangon Biotech (Shanghai, China). Then, the DNA sequence data were confirmed via the GenBank online BLAST software. All the above analysis was to ensure the accuracy of PCR products.

Question 4: In Table 1 and Table 4, unit % is given at the top, so there is no need to duplicate it in the content.

Response 4: Thanks for your comments. We delete % at the top of Table 1 and Table 4 in our revised manuscript.

Question 5: Line 68-69, Please list detailed species that have detected AMR associated with ARGs.

Response 5: Thanks for your comments. We add detailed species (honeybees, poultry and pig) that have detected AMR associated with ARGs in our revised manuscript (Lines 69-70).

Question 6: Please analyze the difference of drug resistance between Dujiang Yan and Barkam in the discussion part.

Response 6: Thanks for your comments. The difference rate of antibiotic resistant between Dujiang Yan and Barkam was added in discussion parts (Lines 230-236): The antibiotic resistance rates of Dujiang yan and Barkam are similar, but the antibiotic resistance rates of Barkam to TE, KZ, CXM, SAM and CIP are significantly higher than Dujiang yan. 

Response to Reviewer 2 Comments

Response to Reviewer:

 Thank you very much for your time and thoughtful comments, many of which have been incorporated into the revised manuscript (Tracked Version). Detailed responses are as follows.

Question 1: In Sample collection, how to know the feces samples from each animal accurately? Please explain methods of collection samples carefully, and to ensure each samples correspondence to each animal.

Response 1: Thanks for your comments. The samples were collected by professional musk deer’s breeder before feeding periods. Breeders can identify musk deer individually by morphological characteristics to ensure that each fecal sample comes from an individual musk deer. As soon as the musk deer defecated, the feces were collected immediately. The process was added in our revised musk deer (line 79-81): Fresh fecal specimens (approximately 10 g) from each musk deer were collected immediately by feeders after defecation on the ground and then quickly transferred into individual 50-mL plastic containers.

Question 2: The results found that there has some difference in AMR/MDR between Dujiang Yan and Barkam, and why this phenomenon was not mentioned in the discussion?

Response 2: Thanks for your comments. Some discussions were added in our revised musk deer (line 232-236): The rate of MDR from Barkam is higher than that from Dujiang Yan, indicateds antibiotic resistant in Bakarm is more severe and more caution should be exercised when choosing antibiotics. The antibiotic resistance rates of Bakarm to TE, KZ, CXM, SAM and CIP are significantly higher than that from Dujiang Yan, which may relate to different living environments

Question 3: In this study, Escherichia coli strains from clinically healthy captive musk deer carrying a large number of ARGs, which make a little difficult to understanding this phenomenon. I suggest in the discussion the author should consider add some information to explain this phenomenon.

Response 3: Thanks for your comments. We have added some explanations and related references to the phenomenon of E. coli carrying ARGs in healthy animals for better understanding by readers in the discussion (lines 238-239): our present study showed E. coli strains from clinically healthy captive non-human primates carrying large number of ARGs (3).

Question 4: Table 8 may not be well expressed, which need revised concise to better understanding. Meanwhile, only one cassette array (dfrA17-aadA5) was identified. Have the results been repeated three times to make sure the accuracy of their results?

Response 4: Thanks for your comments. Table 8 was reorganized in our revised manuscript. As the reviewer mentioned, gene cassette was detected three times in our experiment to make sure the accuracy of our results, and only one cassette array (dfrA17-aadA5) was identified.

Reference:

1. Gorecki A, Musialowski M, Wolacewicz M, Decewicz P, Ferreira C, Vejmelkova D, et al. Development and validation of novel PCR primers for identification of plasmid-mediated colistin resistance (mcr) genes in various environmental settings. Journal of hazardous materials. 2022;425:127936.

2. Rumky J, Kruglova A, Repo E. Fate of antibiotic resistance genes (ARGs) in wastewater treatment plant: Preliminary study on identification before and after ultrasonication. Environmental research. 2022;215(Pt 1):114281.

3. Zhu Z, Jiang S, Qi M, Liu H, Zhang S, Liu H, et al. Prevalence and characterization of antibiotic resistance genes and integrons in Escherichia coli isolates from captive non-human primates of 13 zoos in China. The Science of the total environment. 2021;798:149268.

---

## [Editor Report · Decision Letter 1]

10 Jul 2023

Distribution and Associations for Antimicrobial Resistance and Antibiotic Resistance Genes of Escherichia coli from musk deer (Moschus berezovskii) in Sichuan, China

PONE-D-23-16177R1

Dear Dr. Zhong,

We’re pleased to inform you that your manuscript has been judged scientifically suitable for publication and will be formally accepted for publication once it meets all outstanding technical requirements.

Kind regards,

Prof. Md. Tanvir Rahman, DVM, MSc, PhD

and

Professor of Microbiology,

Department of Microbiology and Hygiene,

Faculty of Veterinary Science,

Bangladesh Agricultural University.

Additional Editor Comments (optional):

Thank you very much for addressing all the comments of the reviewers properly.
---

## [Editor Report · Acceptance letter]

14 Jul 2023

PONE-D-23-16177R1 

Distribution and Associations for Antimicrobial Resistance and Antibiotic Resistance Genes of *Escherichia coli* from musk deer (*Moschus berezovskii*) in Sichuan, China 

Dear Dr. Zhong:

I'm pleased to inform you that your manuscript has been deemed suitable for publication in PLOS ONE. Congratulations! Your manuscript is now with our production department. 

Kind regards, 

on behalf of

Professor Md. Tanvir Rahman 

Academic Editor

PLOS ONE